# Combination of Genomics, Transcriptomics Identifies Candidate Loci Related to Cold Tolerance in Dongxiang Wild Rice

**DOI:** 10.3390/plants11182329

**Published:** 2022-09-06

**Authors:** Dianwen Wang, Yulong Xiao, Hongping Chen, Cheng Huang, Ping Chen, Dazhou Chen, Wei Deng, Jilin Wang

**Affiliations:** Rice National Engineering Research Center (Nanchang), Rice Research Institute, Jiangxi Academy of Agricultural Sciences, Nanchang 330200, China

**Keywords:** Dongxiang wild rice, seedling cold tolerance, quantitative trait locus, transcriptomics, differentially expressed genes

## Abstract

Rice, a cold-sensitive crop, is a staple food for more than 50% of the world’s population. Low temperature severely compromises the growth of rice and challenges China’s food safety. Dongxiang wild rice (DXWR) is the most northerly common wild rice in China and has strong cold tolerance, but the genetic basis of its cold tolerance is still unclear. Here, we report quantitative trait loci (QTLs) analysis for seedling cold tolerance (SCT) using a high-density single nucleotide polymorphism linkage map in the backcross recombinant inbred lines that were derived from a cross of DXWR, and an indica cultivar, GZX49. A total of 10 putative QTLs were identified for SCT under 4 °C cold treatment, each explaining 2.0–6.8% of the phenotypic variation in this population. Furthermore, transcriptome sequencing of DXWR seedlings before and after cold treatment was performed, and 898 and 3413 differentially expressed genes (DEGs) relative to 0 h in cold-tolerant for 4 h and 12 h were identified, respectively. Gene ontology and Kyoto encyclopedia of genes and genomes (KEGG) analysis were performed on these DEGs. Using transcriptome data and genetic linkage analysis, combined with qRT-PCR, sequence comparison, and bioinformatics, *LOC_Os08g04840* was putatively identified as a candidate gene for the major effect locus *qSCT8*. These findings provided insights into the genetic basis of SCT for the improvement of cold stress potential in rice breeding programs.

## 1. Introduction

Rice (*Oryza sativa* L.) is the staple food for half of the world’s population. Increased rice yield has become an important issue for the global economy and food security due to rapid population growth and a reduction in arable land. The growth and development of rice are sensitive to temperature fluctuations, and the optimum temperature for rice cultivation is 25–30 °C [1]. Double-cropping early rice often encounters cold weather in late spring, and suffers from chilling stress during its seedling stage, causing chlorosis, reduction of growth rate and tillering, and low seedling vigor [1,2,3,4]. Thus, improving cold tolerance (CT) in cultivars to promote high and stable rice yield has been a major goal of rice breeding [5].

CT in rice is a quantitative trait that is controlled by quantitative trait loci (QTL) and is largely influenced by the environment. Numerous loci were studied for cold tolerance by QTL mapping and association analysis in rice, including more than 80 QTLs related to seedling cold tolerance (SCT) [4,6,7,8,9,10,11,12,13,14,15,16]. Among these loci, only a few QTLs have been thoroughly researched and cloned using map-based cloning strategies, and most of the functional mechanisms remain largely unknown [16]. *COLD1* encodes a Ca^2+^ signaling regulator that interacts with *qCTS9*, encoding a novel Brassinosteroid Insensitive-1 protein, and co-regulates cold tolerance in rice [17,18]. Functional interaction between *bZIP73* and *bZIP71* can significantly enhance cold tolerance in rice seedlings [19]. The substitution of a single nucleotide at position 343 from A to G in *qPSR10* can effectively enhance cold tolerance in the rice seedling stage [20]. *HAN1* encodes a biologically active jasmonyl-L-isoleucine (JA-Ile) to the inactive form 12-hydroxy-JA-Ile (12OH-JA-Ile) oxidase, which is mediated by fine-tuning JA cold tolerance of rice [21]. *OsWRKY115* (*qCT7*) transcription factor positively regulates the cold tolerance of rice seedlings [22]. In this context, a large effort is still required to explore multiple beneficial alleles for the improvement of cold tolerance in rice.

The development of DNA sequencing technology has supported the possibility of identifying functional genes at the genome-wide level [23,24,25]. RNA-sequencing (RNA-seq), characterizing the genome-wide gene expression profile, is an efficient means to detect stimuli-responsive genes at the genome-wide level [26]. Transcriptome analysis of seedlings before and after cold treatment showed that there are many genes involved in response to cold stress at the seedling stage, such as transport, metabolism, signal transduction, and transcriptional regulation [27,28,29]. In particular, comparative transcriptome analysis of germplasms with distinct cold tolerance phenotypes provides important clues for revealing key genes and mechanisms controlling cold tolerance in rice [30,31,32]. Moreover, the strategy combining transcriptome analysis with QTL mapping was explored to accelerate the identification of candidate genes, including *LOC_Os07g22494*, for a CT-related QTL [33,34,35]. Jiangxi Dongxiang wild rice (DXWR) is the northernmost wild rice in China (28°14′ N), and its seedlings are extremely cold-tolerant [36]. Several QTLs and candidate genes associated with cold tolerance have been identified in DXWR [14,37,38]. However, the genetic basis for the regulation of cold tolerance in DXWR remains unclear.

In this present study, a high-density single nucleotide polymorphism (SNP) linkage map was constructed in the backcross recombinant inbred line (BRIL) population, which was derived from a backcross of DXWR and GZX49 to detect QTLs associated with SCT. At the same time, we cold-treated DXWR seedlings and performed transcriptome sequencing to analyze differentially expressed genes. For a novel major effect, *qSCT8* candidate gene analysis was performed, and *LOC_Os08g04840* encoding an MYB transcription factor was used as the *qSCT8* candidate gene. These findings provide new insights into the genetic basis of SCT in rice.

## 2. Results

### 2.1. Development and Construction of High-Density Bin Map of BRILs

To construct the backcross recombinant inbred lines (BRILs), the F_1_ plant derived from a cross between Ganzaoxian49 (GZX49) and Dongxiang wild rice (DXWR) was backcrossed as a receptor with GZX49. The BC_1_F_1_ seeds were harvested and single-seed descent to obtain a BRIL population comprising 132 lines (Figure 1A). The population was genotyped using the genotyping-by-sequencing (GBS) method, and generated a total of 70.76 Gb of raw data, approximately 536.09 Mb each line, with an overall effective mean depth coverage of 1.97-fold. Based on the Rice Genome Annotation Project (version 7), a total of 55,036 SNPs were detected in the population and were evenly distributed throughout the genome (Figure 1B, Appendix A). According to the high-density genotype of the BRIL population, the recombination breakpoint of each line is determined. In addition, a bin map was constructed based on the recombination breakpoints of each line. The bin map contains 1658 recombination bins distributed on 12 chromosomes, and the average size of the bin is 220.4 kb, with a size range of 12.8 kb to 4.97 Mb. The genetic linkage map was further constructed using a bin map, the total length of the linkage map was 2341.9 cM, and the average interval between adjacent bins was 1.44 cM (Figure 1C, Appendix A).

### 2.2. QTL analysis of Seedling Cold Tolerance

Seedling cold tolerance (SCT) was measured for the two parental lines and the BRILs at the seedling stage. The seedlings grew to trifoliate (Figure 2A), were treated at a low temperature of 4 °C for 84 h in the incubator, and then resumed growth at 28 °C for 7 days; the results showed that the survival rate of DXWR was still 100% after cold treatment, while the survival rate of GZX49 was almost zero (Figure 2B). This indicated that DXWR had strong SCT compared with GZX49. The survival rate of seedlings after cold treatment in the BRIL population showed a continuous distribution from 0 to 100% (Figure 2C,D). These results suggested SCT was in quantitative inheritance controlled by polygenes or QTLs, and there was a large genetic variation for SCT among the BRILs.

The ridge regression analysis for the QTL detection was performed in the BRIL with the bin genotypes. A total of 10 QTLs for SCT were identified and found to explain 48.9% of the phenotypic variance, distributed on all chromosomes except chromosomes 5, 6, and 10 (Figure 3A, Appendix A). The majority (9/10) of the QTLs for SCT had a positive effect, indicating that alleles from DXWR increased SCT (Appendix A). The phenotypic variation explained (PVE) by each QTL ranged from 2.0% to 6.8% (Appendix A), confirming that the SCT was a complex quantitative trait controlled by multiple genes. Among these QTLs, 8 QTLs correspond to bin intervals less than 500 kb, and 5 QTLs correspond to bin intervals that were smaller (less than 200 kb) (Appendix A). The detection of QTLs in relatively small intervals was of great significance for further fine mapping and candidate gene mapping of these QTLs. Based on the mapping interval of two QTLs, *qSCT1* and *qSCT2*, it was found that these two QTLs contained the two reported SCT-related genes *OsMYB3R-2* [39] and *OsTPP1* [40], respectively. Using qRT-PCR technology to detect the expression of *OsMYB3R-2* under 4 °C of treatment, we found that the expression of *OsMYB3R-2* was significantly up-regulated in DXWR but not significantly changed in GZX49 (Figure 3B). Sequence comparison of *OsMYB3R-2* between DXWR and GZX49 revealed in its promoter region (2-kb upstream of the predicted start codon), 17 SNPs and 4 Indels (Figure 3C). Thus, *OsMYB3R-2* was a possible candidate gene for *qSCT1*. Similarly, qRT-PCR technology and sequence comparison were conducted to screen out the degree of response of *OsTPP1* expression to cold stress and the causal polymorphisms of *OsTPP1* (Figure 3D,E). Thus, *OsTPP1* is a possible candidate gene for *qSCT2*. Additionally, these results indicate that the QTLs are entirely effective.

### 2.3. Transcriptome Analysis of Differentially Expressed Genes in DXWR under Cold Stress

To identify the cold stress-responsive genes contained in DXWR, DXWR seedlings were treated at 4 °C, and transcriptome sequencing of aboveground tissues treated for 0 h, 4 h, and 12 h, respectively, was performed. There were 898 and 3413 differentially expressed genes (DEGs) relative to 0 h at 4 °C for 4 h and 12 h, respectively (Figure 4A). Among them, 291 and 607 genes were up- and down-regulated at 4 h, respectively, while 1294 and 2119 genes were up- and down-regulated at 12 h, respectively, by 4 °C treatment (Figure 4A). Additionally, there were 1047 DEGs between 4 h and 12 h of treatment at 4 °C, including 610 up-regulated genes and 437 down-regulated genes (Figure 4A). Among the DEGs at 4 h, 93.4% (273/291) of the up-regulated genes and 86.7% (526/607) of the down-regulated genes were repeatedly detected in the DEGs of 12 h, respectively (Figure 4B,C). In addition, among the differentially expressed genes at 12 h, 79% (1021/1293) of up-regulated genes and 75.2% (1593/2119) of down-regulated genes had no detectable differences at 4 h (Figure 4B,C). This indicated that the response mechanism of DXWR to cold stress was mainly at 12 h of cold treatment.

Furthermore, the DEGs were classified according to gene ontology (GO) and Kyoto encyclopedia of genes and genomes (KEGG) terms. DEGs at 4 h and 12 h were enriched in 14 GO terms, mainly containing mitochondrion, cytosol, chloroplast, plastid, and DNA binding (Figure 4D). KEGG analysis showed that DEGs at 4 h and 12 h displayed enrichment in diverse metabolic pathways, including photosynthesis metabolism, phenylalanine metabolism, starch and sucrose metabolism, amino sugar and nucleotide sugar metabolism, and glycerolipid metabolism (Figure 4E). In addition, the DEGs were classified as plant–pathogen interaction, plant hormone signal transduction, protein processing in the endoplasmic reticulum, and MAPK signaling pathway (Figure 4E).

### 2.4. Candidate Gene Analysis for qSCT8

To identify the possible novel genes for STC within the QTLs, *qSCT8* was thoroughly analyzed because its *p*-value is the most significant. *qSCT8* was first identified within a 638 kb interval. Using Rice Genome Annotation Project (uga.edu), it was found that the interval of *qSCT8* contains about 90 genes. According to the transcriptome sequencing data of DXWR, it was found that among the 90 genes, 18 genes were detected to be expressed at the seedling stage, of which only *LOC_Os08g04840*, encodes an MYB family transcription factor, was up-regulated after cold treatment (Figure 5A). The expression of *LOC_Os08g04840* during 4 °C cold stress in DXWR and GZX49 was further analyzed by qRT-PCR technology, and it was found that the expression of *LOC_Os08g04840* did not change significantly in DXWR and GZX49 after 4 h of treatment compared with 0 h, but it was significantly up-regulated by 2-fold after 12 h of treatment. It was up-regulated by 2-fold, and finally, DXWR was up-regulated by 3.5-fold, and GZX49 was only 2.3-fold after 24 h of treatment (Figure 5B). This indicated that the response intensity of *LOC_Os08g04840* in DXWR was higher than that of GZX49 in the late 4 °C cold treatment. Furthermore, sequence comparison revealed that *LOC_Os08g04840* had substantial variation among rice varieties. In its promoter region (2-kb upstream of the predicted start codon), 4 SNPs were found between DXWR and GZX49. In the coding region, there were no nucleotide variations. Thus, *LOC_Os08g04840* was putatively a possible candidate gene for *qSCT8*.

## 3. Discussion

In recent years, direct seeding of rice has gained popularity due to its time and labor savings and low input. However, the transplanting of seedlings was carried out in a greenhouse, so the genes related to cold tolerance in the seedling stage of rice were lost due to domestication, which leads to major problems such as poor chlorosis, reduced growth rate and tillering, and low seedling vigor in direct-seeded rice when it encounters a cold spring [1,2,41]. Therefore, cultivating rice varieties with strong cold tolerance at the seedling stage is a necessary condition for the application of direct seeding cultivation of rice. DXWR survives heavy snow and can naturally overwinter in its habitat (Figure 6A). The cold tolerance of DXWR is stronger than the main variety LongJing31 in northeast China (Figure 6B). However, most of the identified cold tolerance genes are from japonica rice, which limits the development of cold tolerance breeding in cultivated rice [4,16,18,42,43,44]. Therefore, the identification of SCT-related genes and analysis of the genetic basis of cold tolerance in DXWR has important scientific value for improving the low-temperature seedling vigor ability of rice.

In this study, a high-generation genetic population was constructed using DXWR/GZX49, and a high-density linkage map was constructed using this population, which contained 1658 bins with an average physical interval of 220.4 kb (Appendix A). Using this population, 10 QTLs affecting seedling survival after 78 h of cold treatment at 4 °C were identified, and these QTLs explained 2.06–6.8% of the phenotypic variation range. By comparing the positions of these QTLs, 6 QTLs overlapped with the reported QTLs related to cold tolerance (Appendix A). The location of *qSCT1* was very close to that of the *QTL3* [44], and the identified cold tolerance-related gene *OsMYB3R-2* was included in this mapping interval [39]. The *qSCT2* mapping region contained the identified cold tolerance gene *OsTPP1* [40]. *qSCT4* was mapped near *qCTB4-1*, and two cold tolerance-related genes, *CTB4a* and *CTB2*, have been cloned in this interval [45,46]. *qSCT7* and *qSCT9* contain intervals that overlap with the cloned cold tolerance genes *qCT7* [22] and *qCTS-9* [17], respectively. Two QTL *qSCT11.1* and *qSCT11.2* were identified on chromosome 11, which were detected as *qCTS11-2* and *qCTS11-4*, respectively, in previous reports [43]. The results of the above comparison reflected the accuracy of this study and also illustrated the complexity of genetic regulation of cold tolerance in the rice seedling stage. In addition, this study showed that DXWR was extremely cold tolerant due to its inclusion of multiple cold-tolerant genes. These results not only reinforced those of previous studies but also reflected the complexity of developing strong cold-tolerant rice varieties.

Many rice cold tolerance genes have been identified using reverse genetics, which is important for understanding the regulatory mechanisms of rice cold tolerance [16]. However, only cold tolerance genes with natural allelic variation can be directly used for breeding improvement. Cold stress could induce the up-regulated expression of *OsMYB3R-2*; meanwhile, overexpression of *OsMYB3R-2* in *Oryza sativa japonica* ‘Zhonghua 10’ could significantly enhance the cold tolerance of its seedlings [39]. Low-temperature stress could activate the activity of *OsMAPK3*, and the active *OsMAPK3* could phosphorylate *OsbHLH002* and make it accumulate without ubiquitinase degradation. *OsbHLH002* can promote the expression of *OsTPP1*, thereby increasing trehalose content and resistance to low-temperature damage [40]. Our study found that the *OsMYB3R-2^DXWR^* and *OSTPP1^DXWR^* allele was significantly more up-regulated than *OsMYB3R-2^GZX49^* and *OSTPP1^GZX49^* under cold stress conditions (Figure 3B,D), indicating that *OsMYB3R-2^DXWR^* and *OSTPP1^DXWR^* may be a natural allelic variation with strong cold tolerance, which can be directly used for breeding improvement. This further indicates that DXWR contains many excellent cold tolerance alleles, which can provide important genetic resources for the improvement of rice cold tolerance breeding.

In order to further analyze the cold tolerance characteristics of Dongxiang wild rice at the level of gene expression, transcriptomes were performed on DXWR during cold treatment in this study, and more than 3000 cold stress response genes were identified (Figure 4A–C). GO terms and KEGG pathways analysis were performed on these genes, and it was found that these genes have been involved in multiple biological processes (Figure 4D,E). This further illustrated the complexity of DXWR resistance to cold stress.

Combined QTL-mapping and transcriptomes analysis, compared to the approaches for identifying candidate genes using only traditional QTL-mapping or high-throughput expression profiling, takes less time, reduces labor costs, and increases the selection veracity of the target regions or candidate genes [35,47]. In this study, a novel cold-tolerant QTL, *qSCT8*, was first identified within a 638 kb interval (Appendix A and Figure 3A). The genes *LOC_Os08g04840*, coding an MYB family transcription, that responded to cold stress in the *qSCT8* mapping region were screened out by transcriptome, and qRT-PCR was used to further verify *LOC_Os08g04840* which showed differences in expression before and after cold treatment (Figure 5A,B). In particular, the two G/A variant at the -350 and -354 site upstream of the start codon may cause MADS combined element (AAAAAAAAAGAAAG) defects in *LOC_Os08g04840*^GZX49^ (Figure 5C) [48]. Transcription factors of MYB and MADS family have been reported to regulate cold tolerance [39,49,50,51,52]. The interaction mode of MADS—MYB might play an important role in the signal transmission of cold tolerance in DXWR. This presumption might provide a new idea for the follow-up study on the genetic mechanism of cold tolerance in the DXWR seedling stage.

## 4. Materials and Methods

### 4.1. Plant Materials

A backcross recombinant inbred line (BRIL) population was developed by single-seed descent from a backcross (BC_1_F_1_) of Dongxiang wild rice (DXWR) as donor and *indica* cultivar GZX49 as the recurrent parent. This population consisted of 132 lines, which were backcrossed for 1 generation with GZX49 as the parent, and then selfed for 8 generations (Figure 1A). The BRIL population was grown at the experimental field of Jiangxi Academy of Agricultural Sciences in 2019 at Nanchang (28.57′ N, 115.9′ E), China.

### 4.2. Evaluation of Cold Tolerance at the Seedling Stage

The harvested BRIL seeds were incubated at 40 °C for approximately 36 h to break dormancy and soaked in deionized water at 30 °C for approximately 60 h for germination. A total of 30 germinated seeds from each line were selected and sown in soil in pots and cultivated under a 12 h light/12 h dark cycle at 28 °C/26 °C with 80% humidity. When most of the seedlings grew to trifoliate, weak seedlings were removed, and the rest were treated at a low temperature of 4 °C for 84 h in the incubator and then resumed growth at 28 °C for 7 days. Survival rates were determined after 14-day treatments by counting the surviving plants (leaf contains about 40% green part) and the dead plants. All lines were subjected to three independent replicates.

### 4.3. DNA Extraction and SNP Genotyping

Genomic DNA extraction was carried out using the plant genomic DNA extraction Kit (TIANGEN, Beijing, China), following the manufacturer’s instructions. RNase A was then added to digest RNA. The quality and concentrations of DNA were detected using a NanoDrop 2000 (Thermo Fisher Scientifc, Waltham, MA, USA), while DNA integrity was examined by electrophoresis on 1% agarose gels. To prepare the reduced representation libraries for sequencing, the GBS protocol was carried out according to the method reported by Elshire et al. [53]. In brief, the genomic DNA was first digested by restriction enzymes. In this case, *EcoRI* and *MseI* were selected to effectively reduce genome complexity. Barcode adapters were designed and modified according to the standard Illumina adapter design for paired-end read libraries. The ligation reaction was incubated for 1 h at 22 °C with T4 DNA ligase (Thermo Scientific, Madison, WI, USA) and inactivated at 65 °C for 20 min. The ligation products from each sample were pooled in a single tube, and the products were amplified with 10 cycles of PCR. The amplified library was purified using a QIA quick PCR purification kit (Qiagen, Hilden, Germany), quantified on an Agilent 2100 Bioanalyzer (Agilent Technologies, Palo Alto, CA, USA), and finally sequenced on an Illumina HiSeq3000 instrument (Illumina, San Diego, CA, USA), which generated 150 bp paired-end reads.

The sequencing reads were aligned to the Nipponbare genome sequence (IRGSPv7) using SOAP2 software (version 2.20) [54]. SNP calling was performed with realSFS based on the Bayesian estimation of site frequency at every site. All SNPs were filtered using a Practical Extraction and Report Language (PERL) script based on the following criteria: loci with >50% missing data and minor allele frequency less than 5% in the population. In the end, we obtained 55,036 SNPs. Based on SNP genotyping, the bin was defined by a unique overlapping recombination segment across the BRILs according to a previously reported approach [55]. A bin without breakpoints was generated using the R/qtl package function *fill.geno* with the “argmax” method. The high-density bin map was constructed as previously described [56]. Briefly, the sliding window approach was adopted to evaluate a group of consecutive SNPs for genotyping and determination of recombination breakpoints along the chromosomes of each individual. Blocks with a length less than 250 kb in which the number of sequenced SNPs was fewer than 5 were masked as missing data to avoid false double recombination. Genotypes of bins for regions at the transitions between two different genotype blocks were imputed using the R/qtl package. The genetic linkage map was constructed using the R/qtl package function *est.map* with the Haldane map method [57]. Finally, we used 48,339 SNPs to construct bin maps with 1658 bins. The genotype data of the BRIL population are available on request.

### 4.4. QTL Analysis

The QTL analysis of the phenotypic data with bin maps with 1658 bins in the BILs was performed using the linear ridge regression method to reduce the multicollinearity among markers, as described previously [58]. A significance level of *p* < 0.005 was set as the threshold in the BILs to declare the presence of a putative QTL in a given bin. If several adjacent bins showed *p*-values lower than the threshold, the QTL was tentatively located in the bin (peak bin) with the lowest *p-*value [58]. The phenotypic variance explained by each QTL was decomposed using the “relaimpo” package of R (“lmg” function). QTL nomenclature followed the principles suggested by a previous report [59].

Gene annotations for a given peak bin were obtained from the Rice Genome Annotation Project Database (http://rice.plantbiology.msu.edu/, accessed on 8 June 2021).

### 4.5. RNA-Seq Analysis

Six RNA samples, including cold treatment and controls, were used for RNA sequencing. The aerial parts of 20 seedlings of the corresponding treatment were mixed for each sample in duplicate. In total, 3 μg of RNA per sample was used as the input material for the RNA sample preparations. The sequencing libraries were prepared with the NEBNext Ultra RNA Library Prep Kit for Illumina (NEB, Ipswich, MA, USA) following the manufacturer’s recommendations; index codes were added to attribute sequences to each sample. Clustering of the index-coded samples was performed on a cBot Cluster Generation System with the TruSeq PE Cluster Kit v3-cBot-HS (Illumina) according to the manufacturer’s instructions. After cluster generation, the libraries were sequenced with an Illumina sequencing platform (Hiseq 2500), and 125-bp or 150-bp paired-end reads were generated. The reads were aligned to the reference transcript sequence (http://rice.plantbiology.msu.edu/, accessed on 8 December 2021), and the number of reads covered from the start to the end of each gene was counted. We used the RSEM v1.2.31 tool to quantify gene expression levels. Due to the influence of sequencing depth and gene length, the gene expression value of RNA-seq is generally not represented by read counts, and TPM (Transcripts Per Million mapped reads) is often used as a standardized value. TPM has successively corrected the gene length and sequencing depth. The sum of the TPM of each sample is the same, and the TPM value can reflect the ratio of reads of a gene in the comparison so that the value can be directly compared between samples. Difference analysis was used to find the reasons for the differences in different samples and the degree of their influence on the differences. Statistical analysis of sample expression data was performed to screen genes with significantly different expression levels in different states. The difference analysis is divided into three steps: first, normalize the original read counts, mainly to correct the sequencing depth; then, the statistical model is used to calculate the hypothesis test probability (*p*-value); finally, the multiple hypothesis test correction is performed. Obtain the false discovery rate (FDR) value. For samples with biological replicates, DESeq2 v1.10.1 was selected for differential gene expression analysis. The above RNA-seq-related experiments were completed by Wuhan Genoseq Bioinformatics Technology Co., Ltd. The original RNA-seq data set has been deposited in NCBI (PRJNA871989).

### 4.6. Quantitative Real-Time PCR

Total RNAs were isolated with the TRIzol kit (Invitrogen, Carlsbad, CA, USA) according to the manufacturer’s instructions. The RNA was treated with DNase I (Invitrogen), and approximately 3 μg of total RNA was used to synthesize first-strand cDNA using oligo (dT)_18_ as a primer (Promega, Shanghai, China). The trans-intron ACTIN primer (ACTIN-M) was used to detect whether the reverse transcribed cDNA still has genomic DNA, and the cDNA without genomic DNA was used for subsequent Quantitative real-time PCR. Quantitative real-time (qRT) PCR was performed using gene-specific primers and the FastStart Universal SYBR Green Master (Roche) on a real-time PCR ViiA7 system (Applied Biosystems). Genes *ubiquitin* (*LOC_Os03g13170*) and *actin1* (*LOC_Os03g50885*) not differentially expressed in RNAseq were used as the internal control. The relative quantification method (2^−^^△△^^CT^) was used to evaluate gene expression level [60]. As similar expression results were observed regardless of which control genes were used, the *ubiquitin* of expressions was applied for the relative expression analyses for every assayed sample. At least three biological replicates, each containing four technical, were performed for each experiment.

The primers were designed according to the Nipponbare reference genome by Primer3 (http://redb.ncpgr.cn/modules/redbtools/primer3.php, accessed on 8 April 2022). The sequences were analyzed using Sequencer 5.0 (Gene Codes Corporation, Ann Arbor, MI, USA). All primers were synthesized at Sangon Biotech (Shanghai, China) and are listed in Appendix A.

## 5. Conclusions

In conclusion, Dongxiang wild rice (DXWR) has strong cold tolerance. A BRIL population of DXWR/GZX49 with a high-density bin map was used to identify several QTLs and potential candidate genes for seedling cold tolerance in this study. Cold stress-responsive genes in DXWR were detected by transcriptome sequencing. Moreover, using transcriptome data and genetic linkage analysis, combined with qRT-PCR, sequence comparison, and bioinformatics, the candidate gene of the major effect locus *qSCT8* was identified as *LOC_Os08g04840*. These findings might provide insights into the genetic basis of SCT for the improvement of cold stress potential in rice breeding programs.

## Figures and Tables

**Figure 1 plants-11-02329-f001:**
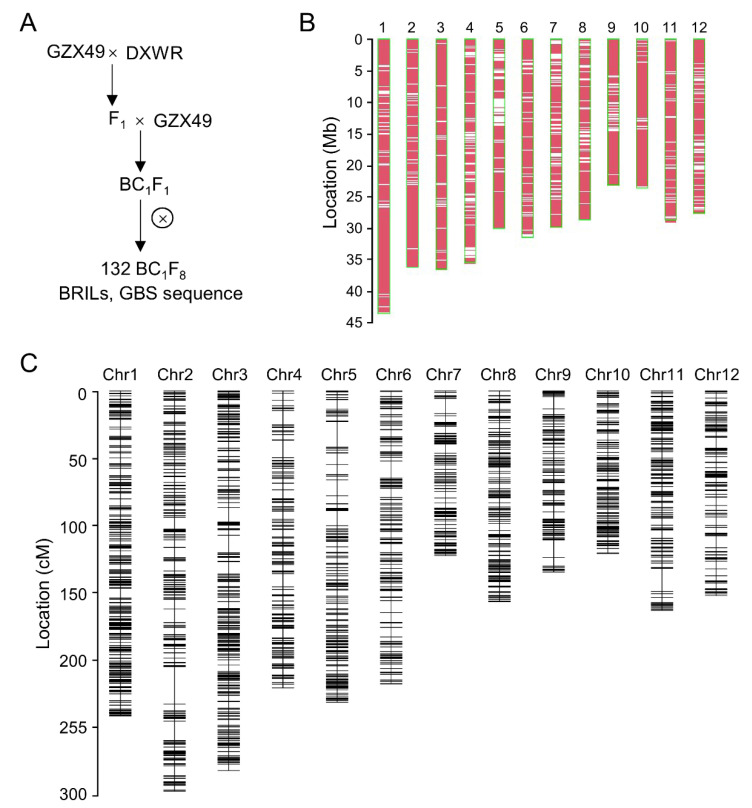
Development and construction of high-density bin map of BRILs. (**A**) Flowchart of developing backcross recombinant inbred line (BRIL) population. ⊗ represents self-cross. (**B**) Polymorphic SNPs between DXWR and GZX49 distributed on chromosomes. (**C**) Genetic linkage map constructed by population BRILs.

**Figure 2 plants-11-02329-f002:**
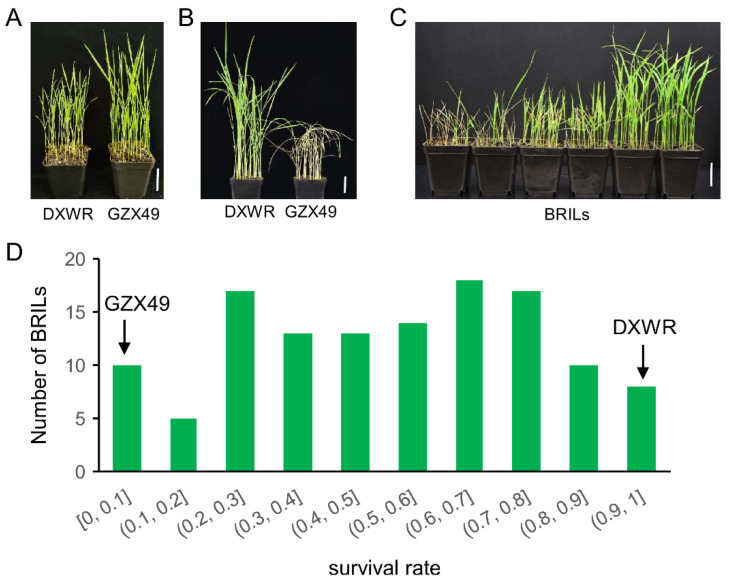
Phenotypic identification of seedling for cold tolerance. (**A**,**B**) represent the phenotypes of GZX49 and DXWR before and after cold treatment, respectively. (**C**) Phenotypes of some lines of the BRIL population after cold treatment. The scale bar of (**A**–**C**) means 5 cm. (**D**) Frequency distribution of seedling survival rate after cold treatment. Arrows indicate the means of parental lines GZX49 and DXWR.

**Figure 3 plants-11-02329-f003:**
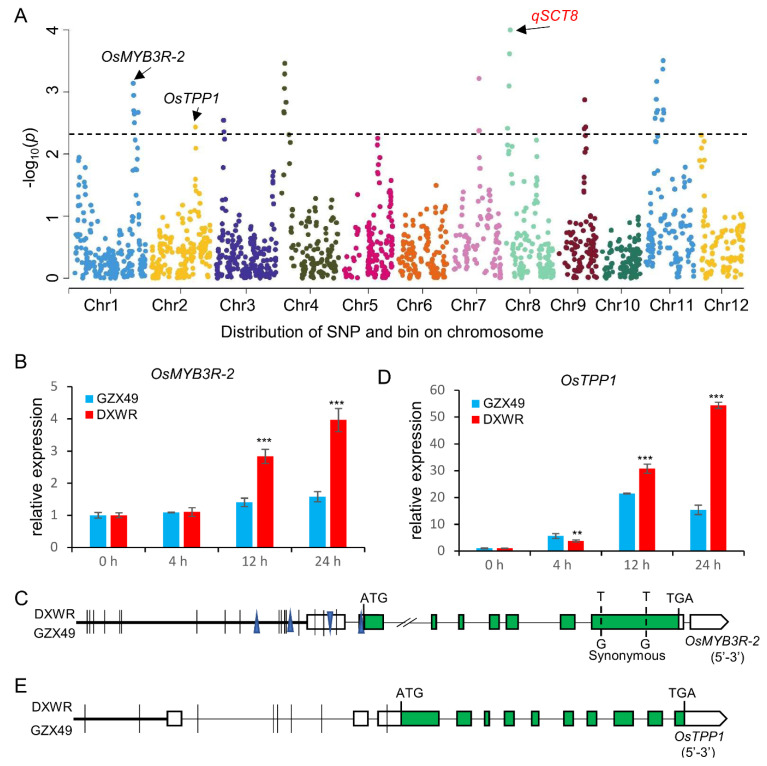
QTL analysis of the BRIL population. (**A**) Manhattan plots of the loci for seedling cold tolerance (SCT). The *x*-axis represents single nucleotide polymorphism (SNP) along each numbered chromosome; the *y*-axis represents the negative logarithm of the *p*-value (-log10 p) for the SNP association. Horizontal dashed lines in the plots indicate the declaration thresholds. (**B**,**D**) Expression levels of the *OsMYB3R-2* and *OsTPP1* in GZX49 and DXWR after cold stress measured by qRT-PCR, respectively. The results were statistically analyzed using Student’s *t*-test (** *p *< 0.01, *** *p* < 0.005). Transcription levels relative to 0 h, which was set to 1, are presented as the mean and SE of triplicates. *LOC_Os03g13170* (Ubiquitin) is the control gene. (**C**,**E**) Sequence comparison of *OsMYB3R-2* and *OsTPP1* among GZX49 and DXWR. The vertical bars and triangle represent SNPs and nucleotide deletion, respectively.

**Figure 4 plants-11-02329-f004:**
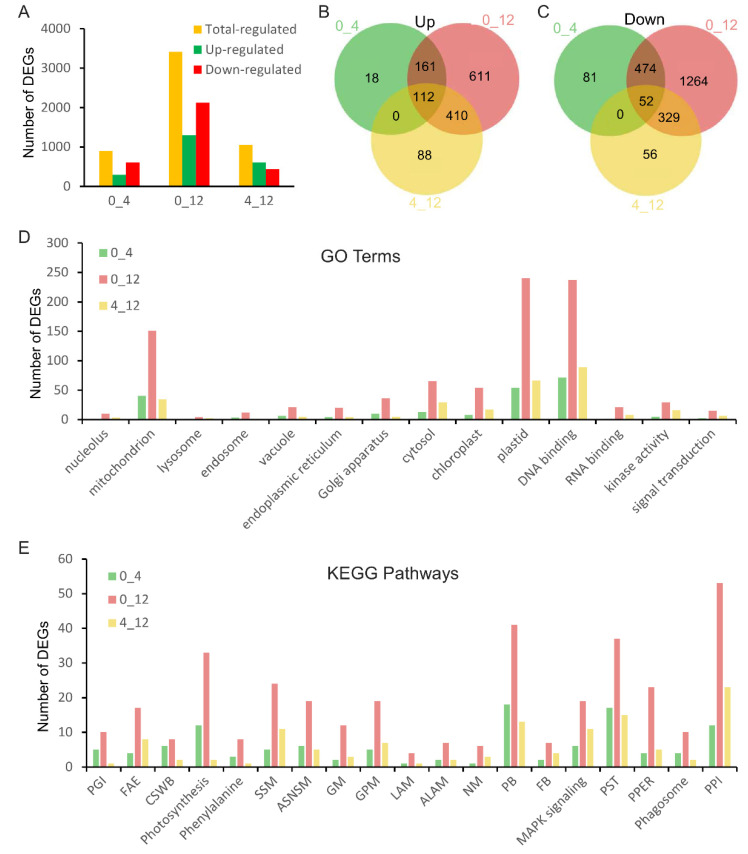
Transcriptome analysis of the genetic mechanism of DXWR in response to cold stress. (**A**) Differentially expressed gene statistics. (**B**,**C**) are Venn diagrams of up-regulated and down-regulated genes, respectively. (**D**,**E**) are gene ontology (GO) functional enrichment histogram and Kyoto encyclopedia of genes and genomes (KEGG) pathway enrichment histogram, respectively. 0_4, 0_12, and 4_12 represent the comparison of DXWR seedling treatment at 4 °C for 0 h and 4 h, 0 h and 12 h, and 4 h and 12 h, respectively. PGI, FAE, CSWB, SSM, ASNSM, GM, GPM, LAM, ALAM, NM, PB, FB, PST, PPER, and PPI in subfigure (**E**) represent pentose and glucuronate interconversions, fatty acid elongation, cutin suberine and wax biosynthesis, starch and sucrose metabolism, amino sugar and nucleotide sugar metabolism, glycerolipid metabolism, gycerophospholipid metabolism, linoleic acid metabolism, alpha-linolenic acid metabolism, nitrogen metabolism, phenylpropanoid biosynthesis, flavonoid biosynthesis, plant hormone signal transduction, protein processing in endoplasmic reticulum, and plant–pathogen interaction, respectively.

**Figure 5 plants-11-02329-f005:**
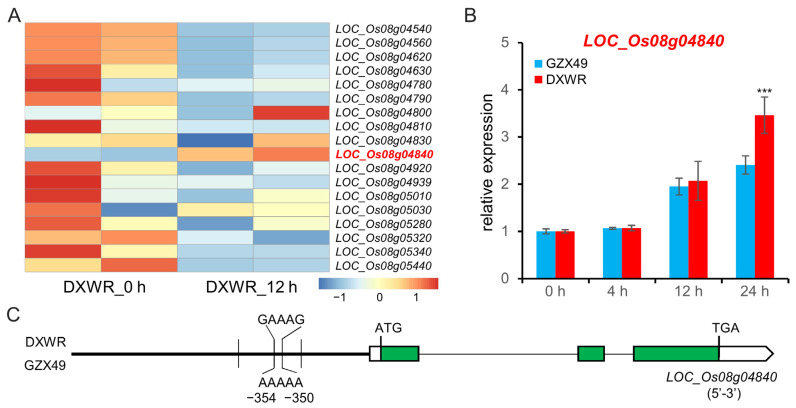
*qSCT8* candidate gene analysis. (**A**) Clustering heat maps of the relative expression levels of genes within the qSCT8 localization interval determined using RNA-seq data. Standard scores (Z-scores) were used as the numerical signs to evaluate the standard deviations from the mean of the corresponding samples. (**B**) Expression levels of the *LOC_Os08g04840* in GZX49 and DXWR after cold stress measured by qRT-PCR. The results were statistically analyzed using Student’s *t*-test (*** *p*  <  0.005). (**C**) Sequence comparison of *LOC_Os08g04840* among GZX49 and DXWR. Vertical bars represent SNPs.

**Figure 6 plants-11-02329-f006:**
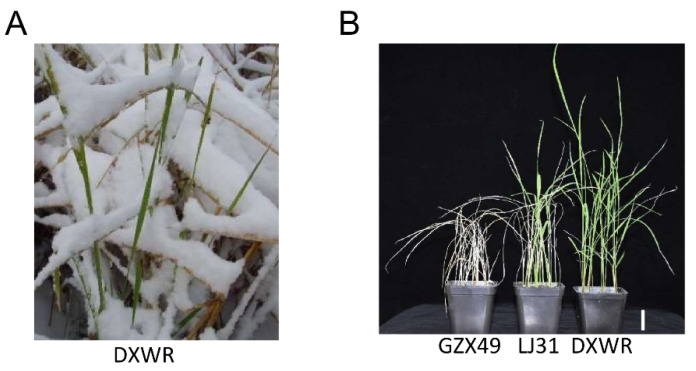
Cold tolerance phenotype of DXWR. (**A**) The phenotype of DXWR encountering heavy snow weather in its habitat. (**B**) Phenotypes of DXWR, Longjing 31, and GZX49 after 4 days of treatment at 0 °C at the seedling stage. LJ31 stands for LongJing31. The scale bar of (**B**) means 5 cm.

## Data Availability

Data is contained within the article and Appendix A.

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
