# Peer review of "Combination of Genomics, Transcriptomics Identifies Candidate Loci Related to Cold Tolerance in Dongxiang Wild Rice"

_plants, 2022, doi:10.3390/plants11182329_

Round 1

Reviewer 1 Report

The manuscript is overall well-written and well presented, revealing an interesting and an integrated approach, using several molecular techniques, to studying cold tolerance of Dongxiang wild rice. It was very easy to read and data have been explored well. However, several important issues have to be revised before its consideration to publication, namely at Material and Methods section. Also, all sequencing gata generated from SNP genotyping and RNA seq must be deposited at an available repository.

Specifically:

Sub-section- DNA extraction and SNP genotyping- How was the  gDNA libraries built? how many samples were used? This important information on sampling has to be included.

Sub-section RNA seq-analysis- How many samples were used? How many biological replicates were perfoemed? include this important information. Also, no text is devoted to RNA seq bioinformatic analysis and statistics. Please include this data.

Sub-section- Quantitative Real-Time PCR- Several informations on methods have be clarified. Did you performed the test for gDNA detection after DNAse treatment? this step is important, as gDNA, is very difficult to remove and should be tested prior to cDNA synthesis.

Again, it is missing how many saples were used, how many biological replicates, etc..

Besides, the use of only one reference gene is not advisable, since you have to test if no changes have occurred under your treatment conditions. Considering the RNA seq approach done, why not selecting at least 2 genes that have not been regulated upon the tretaments tested? have you used the same RG for the treatments tested? Also, melting curves from all genes (RGs and genes tested) sould be included.

Reviewer 2 Report

The authors put a lot of effort into this study by generating 132 NILs developed by backcrossing cold-tolerant wild parent with a high-yielding recurrent parent. The obtained lines were genotyped using an abundant number of SNPs, high-density linkage map was constructed and QTL analysis was performed. In addition, gene expression, transcriptome sequencing, and bioinformatic analysis were applied to explore candidate genes for cold tolerance in rice. Accordingly, the manuscript provides a fairly robust dataset and deserves to be published after minor revision.  

Specific comments

The manuscript needs major English editing.

The term “recombinant inbred lines” is usually used for lines in an advanced generation that were generated from bi-parent and selfed for various cycles. I think “near-isogenic lines” is more appropriate for lines developed by the backcrossing method as in your case in this study.

The introduction section needs to be revised and highlight the importance of genomics and transcriptomics tools in identifying candidate loci related to cold tolerance in wild plants. The hypothesis and knowledge gap need to be clarified. 

The materials and methods section needs to be well revised and complete the missing data. For example, under the subtitle “Plant materials” in lines 283-287, the number of generated lines should be clarified. Besides, the number of backcrossing with recurrent parent and the number of selfing generations should be presented. Moreover, under the subtitle “DNA Extraction and SNP Genotyping” the number of all applied SNPs should be described as well as the excluded number and the used number for the QTL analysis.

Reference number 37, 38, and 39 and their corresponding discussed findings in the results section should be moved to the discussion section.

The discussion section is very short, does not cover all obtained results, and accordingly, needs to be extended and improved.

Scientific names should be in italic throughout the manuscript as lines 406,411, 413, 461,462 …..468, 483

Please standardize the reference style, as the abbreviated journal name followed by a full stop or not.

Round 2

Reviewer 1 Report

The manuscript has been improved significantly by the authors addressing positively to comments.

Minor comments: on the qPCR methods, include how was the PCR efficiency reaction. Also, the melting curves should be included as supplementary data and in table S3 add the melting curves temperatures and also qPCR efficiency percentage values.